# Efficient inference for time-varying behavior during learning

**Nicholas A. Roy**[1]  **Ji Hyun Bak**[2]  **Athena Akrami**[1,3,*]

**Carlos D. Brody**[1,3,4]  **Jonathan W. Pillow**[1,5]

[1]Princeton Neuroscience Institute, Princeton University
[2]Korea Institute for Advanced Study    [3]Howard Hughes Medical Institute
[4]Dept. of Molecular Biology, [5]Dept. of Psychology, Princeton University
[*]*current address at* Sainsbury Wellcome Centre, UCL
{nroy,brody,pillow}@princeton.edu,
jhbak@kias.re.kr, athena.akrami@ucl.ac.uk

## Abstract

The process of learning new behaviors over time is a problem of great interest in both neuroscience and artificial intelligence. However, most standard analyses of animal training data either treat behavior as fixed or track only coarse performance statistics (e.g., accuracy, bias), providing limited insight into the evolution of the policies governing behavior. To overcome these limitations, we propose a dynamic psychophysical model that efficiently tracks trial-to-trial changes in behavior over the course of training. Our model consists of a dynamic logistic regression model, parametrized by a set of time-varying weights that express dependence on sensory stimuli as well as task-irrelevant covariates, such as stimulus, choice, and answer history. Our implementation scales to large behavioral datasets, allowing us to infer 500K parameters (e.g., 10 weights over 50K trials) in minutes on a desktop computer. We optimize hyperparameters governing how rapidly each weight evolves over time using the decoupled Laplace approximation, an efficient method for maximizing marginal likelihood in non-conjugate models. To illustrate performance, we apply our method to psychophysical data from both rats and human subjects learning a delayed sensory discrimination task. The model successfully tracks the psychophysical weights of rats over the course of training, capturing day-to-day and trial-to-trial fluctuations that underlie changes in performance, choice bias, and dependencies on task history. Finally, we investigate why rats frequently make mistakes on easy trials, and suggest that apparent lapses can be explained by sub-optimal weighting of known task covariates.

## 1   Introduction

A vast swath of modern neuroscience research requires training animals to perform specific tasks. This training is expensive and time-consuming, yet the data collected during the training period are often discarded from analysis. Moreover, animals can learn at vastly different rates, and may learn different strategies to achieve a criterion level of performance in a given task. Most neuroscience studies ignore such variability, and commonly track only coarse statistics like accuracy and bias during training. These statistics are not sufficient to reveal subtle differences in strategy, such as unequal weighting of task variables or reliance on particular aspects of trial history. However, behavior collected during training may provide valuable insights into an animal's mental arsenal of problem solving strategies,

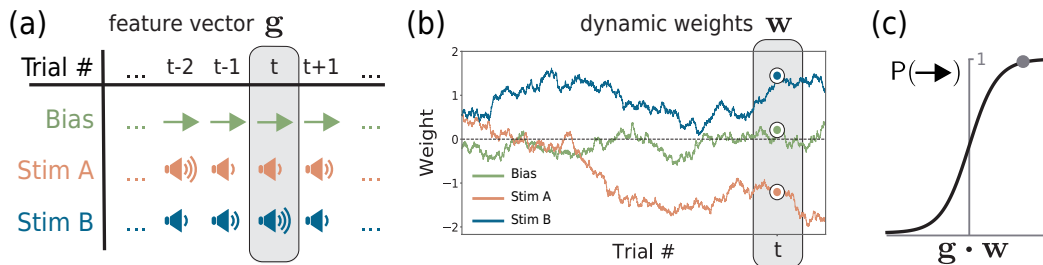

Figure 1: Model schematic. **(a)** On each trial, a variety of both task-related and task-irrelevant variables may affect an animal's choice behavior. We call the carrier vector of all $K$ input variables on a particular trial $\mathbf{g}_t$. **(b)** As the animal trains on the task, psychophysical weights $\mathbf{w}_t$ evolve with independent Gaussian noise, altering how strongly each variable affects behavior. **(c)** The probability of "right" given the input is described by a logistic function of $\mathbf{g}_t \cdot \mathbf{w}_t$.

and uncover how those strategies evolve with experience. Understanding detailed differences in behavior may shed light on differences in neural activity across animals or task conditions, reveal general aspects of behavior, or inspire the development of new learning algorithms [1].

One reason training data are frequently ignored is a lack of good methods for tracking behavior during training, or for tracking continued learning after the dedicated training phase has ended. Of the few approaches to characterizing time-varying psychophysical behavior, perhaps the simplest is to apply standard logistic regression to separate blocks of trials. While useful in certain specific situations, there are numerous drawback to such a blocking approach, including: the need to choose a block size, the removal of dependencies between adjacent blocks, and the inability to track finer time-scale changes within a block. Such an approach also assumes that there is a single timescale at which all psychophysical weights vary. Smith and Brown introduced an assumed density filtering method for tracking psychophysical performance on a trial-by-trial basis [2]. This approach of explicitly tracking a parameter to determine the earliest time at which statistically significant learning had occurred during training has been extended in various contexts [3, 4]. Here we propose an alternate approach based on exact MAP estimation of time-varying psychophysical weights, with efficient and scalable methods for inferring hyperparameters governing the timescale of changes for different weights.

In this paper, we present a dynamic logistic regression model for time-varying psychophysical behavior. Our model quantifies animal behavior at single-trial resolution, allowing for intuitive visualization of learning dynamics and direct analysis of psychophysical weight trajectories. We develop efficient inference methods that exploit sparse structure in order to scale to large datasets with high-dimensional, time-varying psychophysical weights. Moreover, we use the decoupled Laplace approximation method [5] to perform highly efficient approximate maximum marginal likelihood inference for a set of hyperparameters governing the rates of change for different psychophysical weights. We apply our method to a large behavioral data set of rats demonstrating a variety of constantly evolving complex behaviors over tens of thousands of trials, as well as human subjects with significantly more stable behavior. We compare the predictions of our model to conventional measures of behavior, and conclude with an analysis of lapses on perceptually easy trials to demonstrate the model's explanatory power. We expect that our method will provide immediate practical benefit to trainers, in addition to giving unprecedented insight into the development of new behaviors. An implementation of all methods are available as the Python package PsyTrack [6].

## 2   Dynamic logistic regression model

Here we describe our dynamic model for time-varying psychophysical behavior. We consider a general two-alternative forced choice (2AFC) sensory discrimination task in which the animal is presented with a stimulus $\mathbf{x}_t \in \mathbb{R}^d$, and makes a choice $y_t \in \{0, 1\}$ between two options that we will refer to as "left" and "right" (although the method can be extended to multi-choice tasks [7]).

We model the animal's behavior as depending on an internal model parametrized by a set of weights $\mathbf{w}_t \in \mathbb{R}^K$ that govern how the animal's choice depends on an input "carrier" vector $\mathbf{g}_t \in \mathbb{R}^K$ for the current trial $t$ (Fig. 1a,b). This carrier $\mathbf{g}_t$ contains the task stimuli $\mathbf{x}_t$ for the current trial, as well as a

variety of additional covariates (e.g., stimulus, choice, or answer history over the preceding one or more trials), and a constant "1" to capture bias towards one choice or the other (see Sec. S1 for more detail). Empirically, animal behavior in early training often exhibits dependencies on both stimulus and choice history [8–10]; including these features is therefore critical to building an accurate model of the animal's evolving psychophysical strategy (we return to this with a study of lapses in Sec. 6).

Given the weight and carrier vectors, the animal's choice behavior on a given trial is described by a Bernoulli generalized linear model (GLM), also known as the logistic regression model (Fig. 1c):

$$p(y_t \mid \mathbf{g}_t, \mathbf{w}_t) = \frac{\exp(y_t(\mathbf{g}_t \cdot \mathbf{w}_t))}{1 + \exp(\mathbf{g}_t \cdot \mathbf{w}_t)}. \tag{1}$$

Unlike standard psychophysical models, which assume weights are constant across trials and that behavior is therefore constant, we instead assume that the weights evolve gradually through time. We model this evolution with independent Gaussian innovations noise added to the weights after each trial [11, 12]:

$$\mathbf{w}_t = \mathbf{w}_{t-1} + \boldsymbol{\eta}_t, \quad \boldsymbol{\eta}_t \sim \mathcal{N}(0, \operatorname{diag}(\sigma_1^2, \ldots, \sigma_K^2)), \tag{2}$$

where $\mathbf{w}_t$ denotes the weight vector on trial $t$, $\boldsymbol{\eta}_t$ is the noise added to the weights after the previous trial, and $\sigma_k^2$ denotes the variance of the noise for weight $k$, also known as the *volatility* hyperparameter. Here $\operatorname{diag}(\sigma_1^2, \ldots, \sigma_K^2)$ denotes a diagonal matrix with the volatility hyperparameters for each weight along the main diagonal. We note that this choice of prior on $\mathbf{w}$ is largely agnostic, though more structured priors could be considered.

## 3 Inference

Inference involves fitting the entire trajectory of the weights from the noisy response data collected over the course of experiment. This amounts to a very high-dimensional optimization problem when we consider models with several weights and datasets with tens of thousands of trials. Moreover, we wish to learn the volatility hyperparameters $\sigma_k$ in order to determine how quickly each weight evolves across trials.

### 3.1 Efficient global optimization for $\mathbf{w}_{\mathrm{MAP}}$

Let $\mathbf{w}$ denote the massive weight vector formed by concatenating all of the length-$N$ trajectory vectors for each weight $k = 1, \ldots, K$, where $N$ is the total number of trials. We can then express the prior over the weights by noting that $\boldsymbol{\eta} = D\mathbf{w}$, where $D$ is a block-diagonal matrix of $K$ identical $N \times N$ difference matrices (i.e., 1 on the diagonal and $-1$ on the lower off-diagonal). Because the prior on $\boldsymbol{\eta}$ is simply $\mathcal{N}(\mathbf{0}, \Sigma)$, where $\Sigma$ has each of the $\sigma_k^2$ stacked $N$ times along the diagonal, the prior for $\mathbf{w}$ is $\mathcal{N}(\mathbf{0}, C)$ with $C^{-1} = D^{\top}\Sigma^{-1}D$. The log-posterior is then given by

$$\log p(\mathbf{w}|\mathcal{D}) = \tfrac{1}{2}(\log|C^{-1}| - \mathbf{w}^{\top}C^{-1}\mathbf{w}) + \sum_{t=1}^{N} \log p(y_t|\mathbf{g}_t, \mathbf{w}_t) + const, \tag{3}$$

where $\mathcal{D} = \{(\mathbf{g}_t, y_t)\}_{t=1}^{N}$ is the set of input carriers and responses, and $const$ is independent of $\mathbf{w}$.

Our goal is to find the $\mathbf{w}$ that maximizes this log-posterior. With $NK$ total parameters (potentially 100's of thousands) in $\mathbf{w}$, however, most procedures that perform a global optimization of all parameters at once are not feasible; for example, related work has calculated trajectories by maximizing the likelihood using local approximations [2]. Whereas the use of the Hessian matrix for second-order methods often provides dramatic speed-ups, a Hessian of $(NK)^2$ parameters is usually too large to fit in memory (let alone invert) for $N > 1000$ trials. On the other hand, we observe that the Hessian of our log-posterior is sparse:

$$H = \frac{\partial^2}{\partial \mathbf{w}^2} \log p(\mathbf{w}|\mathcal{D}) = C^{-1} + \frac{\partial^2 L}{\partial \mathbf{w}^2} \tag{4}$$

where $C^{-1}$ is a sparse (banded) matrix, and $\partial^2 L / \partial \mathbf{w}^2$ is a block-diagonal matrix. The block diagonal structure arises because the log-likelihood is additive over trials, and weights at one trial $t$ do not affect the log-likelihood component from another trial $t'$. We take advantage of this sparsity, using a variant of conjugate gradient optimization that only requires a function for computing the product of the Hessian matrix with an arbitrary vector [13]. Since we can compute such a product using only sparse terms and sparse operations, we can utilize quasi-Newton optimization methods in SciPy to find a global optimum for our weights, even for very large $N$ [14].

**Algorithm 1** Optimizing hyperparameters with the *decoupled* Laplace approximation

---
**Require:** input carriers $\mathbf{g}$, choices $\mathbf{y}$
**Require:** initial hyperparameters $\theta_0$, subset of hyperparameters to be optimized $\theta_{\text{OPT}}$
 1: **repeat**
 2:     Optimize for $\mathbf{w}$ given current $\theta \longrightarrow \mathbf{w}_{\text{MAP}}$, Hessian of log-posterior $H_\theta$, log-evidence $E$
 3:     Determine Gaussian prior $\mathcal{N}(\mathbf{0}, C_\theta)$ and Laplace appx. posterior $\mathcal{N}(\mathbf{w}_{\text{MAP}}, -H_\theta^{-1})$
 4:     Calculate Gaussian approximation to likelihood $\mathcal{N}(\mathbf{w}_L, \Gamma)$ using product identity, where
        $\Gamma^{-1} = -(H_\theta + C_\theta^{-1})$ and $\mathbf{w}_L = -\Gamma H_\theta \mathbf{w}_{\text{MAP}}$
 5:     Optimize $E$ w.r.t. $\theta_{\text{OPT}}$ using closed form update (with sparse operations)
        $\mathbf{w}_{\text{MAP}} = -H_\theta^{-1} \Gamma^{-1} \mathbf{w}_L$
 6:     Update best $\theta$ and corresponding best $E$
 7: **until** $\theta$ converges
 8: **return** $\mathbf{w}_{\text{MAP}}$ and $\theta$ with best $E$

---

## 3.2  Hyperparameter fitting with the decoupled Laplace approximation

So far we have addressed the problem of finding a global optimum for $\mathbf{w}$ given a specific hyperparameter setting $\theta = \{\sigma_k\}$; now we must also find the optimal hyperparameters. Cross-validation is not easily applied given the number of different volatility parameters, and so we turn instead to approximate marginal likelihood. To select between models with different $\theta$, we use a Laplace approximation to the posterior, $p(\mathbf{w}|\mathcal{D}, \theta) \approx \mathcal{N}(\mathbf{w}|\mathbf{w}_{\text{MAP}}, -H^{-1})$, to estimate the marginal likelihood (or evidence) as [15]:

$$p(\mathbf{y}|\mathbf{g}, \theta) = \frac{p(\mathbf{y}|\mathbf{g}, \mathbf{w})\, p(\mathbf{w}|\theta)}{p(\mathbf{w}|\mathcal{D}, \theta)} \approx \frac{\exp(L) \cdot \mathcal{N}(\mathbf{w}|\mathbf{0}, C)}{\mathcal{N}(\mathbf{w}|\mathbf{w}_{\text{MAP}}, -H^{-1})}. \tag{5}$$

Naive optimization of $\theta$ requires a re-optimization of $\mathbf{w}_{\text{MAP}}$ for every change in $\theta$, strongly restricting the dimensionality of tractable $\theta$ to whatever could be explored with grid search; the simplest approach is to reduce all $\sigma_k$ to a single $\sigma$, as assumed in [16].

Here we use the decoupled Laplace method [5] to avoid the need to re-optimize for our weight parameters after every update to our hyperparameters by making a Gaussian approximation to the likelihood of our model. The optimization is explained in Algorithm 1. By circumventing nested optimization of $\theta$ and $\mathbf{w}$, we can consider larger sets of hyperparameters and more complex priors over our weights, while still fitting in minutes on a laptop (Fig. 2c). For example, letting each weight evolve with its own distinct $\sigma_k$ often allows for both a more accurate fit to data and additional insight into the dynamics (as in Fig. 3b). In practice, we also parametrize $\theta$ by fixing $\sigma_{k,t=0} = 16$, an arbitrary large value that allows the likelihood to determine $\mathbf{w}_0$ rather than forcing the weights to initialize near some predetermined value.

## 3.3  Overnight dynamics

Another specific parametrization of $\theta$ made possible by the decoupled Laplace method is the inclusion of an additional type of hyperparameter, $\sigma_{\text{day}}$, to modulate the change in weights occurring between training sessions. Intuitively, one might expect that between the last trial of a session and the first trial of the next session, change in behavior is greater than between trials that are consecutive within the same session. By indexing the first trial of each session, we can introduce a new set of hyperparameters $\{\sigma_{k,\text{day}}\}$ which we can then optimize to account for the between-session changes within each weight.

Whereas all $2 \cdot K$ hyperparameters in $\theta = \{\sigma_1, \ldots, \sigma_K, \sigma_{1,\text{day}}, \ldots, \sigma_{K,\text{day}}\}$ can have distinct values in the most flexible version of the model, there are certain optional constraints that may be more relevant to animal behavior. For example, when both the $\{\sigma_k\}$ and $\{\sigma_{k,\text{day}}\}$ are fixed to be very small, it means that weights effectively do not change, replicating the standard logistic regression model with constant weights. On the other hand, when fixing $\{\sigma_k\}$ to be very small and $\{\sigma_{k,\text{day}}\}$ to be very large, we would recover a different set of constant weights for each session, replicating a particular blocked approach to logistic regression discussed earlier. By only fixing the $\{\sigma_{k,\text{day}}\}$ to be large while optimizing freely over each of $\{\sigma_k\}$, we essentially find the best weight *trajectory* within each session, while allowing the weights to "reset" at the start of each new session. The *decoupled* Laplace

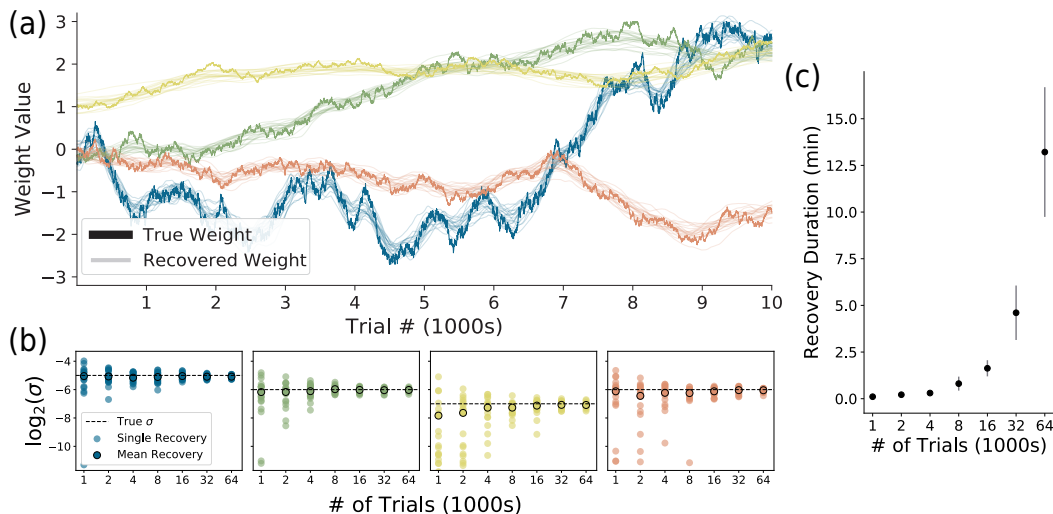

Figure 2: Recovering weights and hyperparameters from simulated data. **(a)** Generating 20 behavioral realizations ($\mathbf{y}$'s) from one simulated set of $K\!=\!4$ weights (in bold), we recover 20 sets of weights (faded). Observe that the recovered weights closely track the real weights in all realizations. **(b)** The hyperparameters $\sigma_k$ recovered for each weight over 20 distinct simulations, as a function of number of trials. Note that with more trials, the recovered $\sigma_k$ converge to the true $\sigma_k$ (dotted black line). **(c)** Average computation time for full optimization of weights and hyperparameters for a single realization. Even with tens of thousands of trials, this model can be fit in minutes on a laptop.

method makes it feasible to optimize over any subset of these hyperparameters at once, allowing exploration of many types of models and the localization of behavioral dynamics to specific weights or periods of training.

## 4 Simulation results

We first demonstrate our method using simulated data. We generate $K\!=\!4$ weight trajectories over $N\!=\!64,000$ trials, simulating each as a Gaussian random walk with variance $\sigma_k^2$ and a reflecting boundary at $\pm 4$. For each trial, we then drew the carrier vector $\mathbf{g}_t$ from a standard normal, calculated $P(\text{Right})$, and used this probability to sample a choice $y_t$. Since our model is probabilistic (Eq. 1), we can draw many behavioral realizations ($\mathbf{y}$'s) from the same "true" weight trajectories. Our method not only accurately estimates the weight trajectories across realizations (Fig. 2a), but also recovers the hyperparameter $\sigma_k$ for each weight across many different simulations (Fig. 2b). We also tested the scalability of the method over increasing number of trials (Fig. 2c). We note that having more than 64K trials for a single animal is highly unusual, and so fifteen minutes of computation time on a laptop is a rough upper bound for most practical use; behavioral datasets commonly have only a few thousand trials and can be fit in seconds. In order to confirm the efficacy of our *decoupled* Laplace method in recovering the best setting of hyperparameters, we confirm with grid search that the algorithm converges on the hyperparameters with the highest evidence and highest cross-validated log-likelihood on simulated data (see Fig. S1).

## 5 Behavioral dynamics in rats & humans

To further explore the advantages and insights provided by our model, we apply our method to behavioral data from both rats and humans performing a 2AFC delayed response task, as reported in [17]. The task involves the presentation of two auditory stimuli of different amplitude, separated by a delay. If the first stimulus (Tone A) is louder than the second (Tone B), then the subject must go right to receive a reward, and vice-versa (Fig. 3a; for more detail see [17]). In our model, the "correct" set of weights for performing this task with high accuracy are a large, positive weight for Tone A, an equal and opposite weight for Tone B (the two *sensitivities* to stimuli), and zeros for all other (task-irrelevant) weights. We applied our method to early training data from 20 rats and 9 human

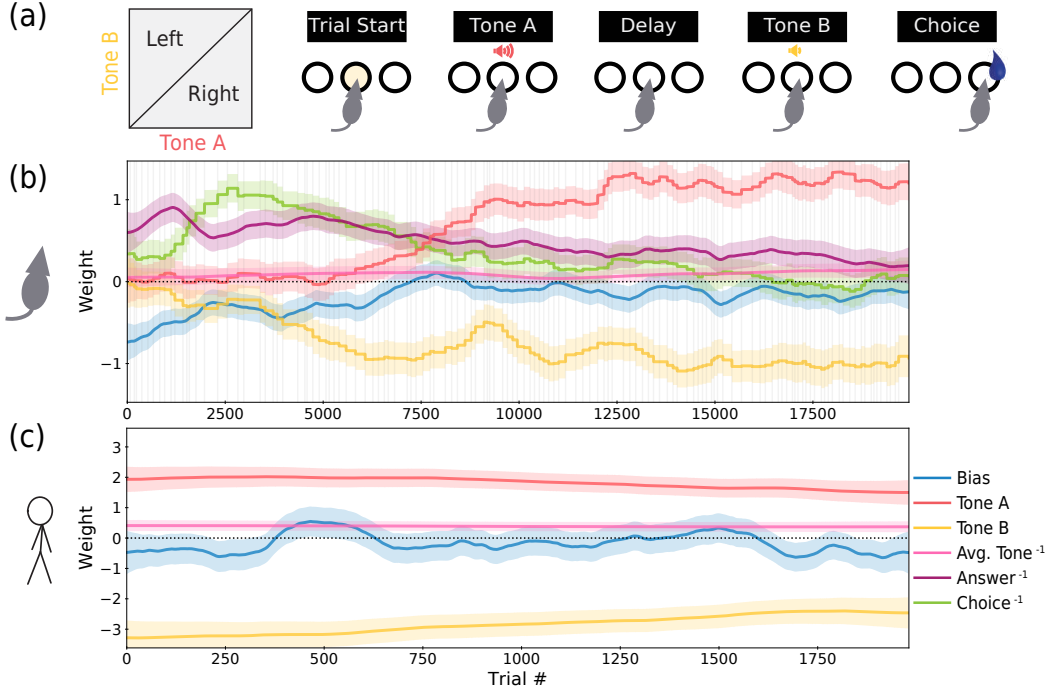

Figure 3: Application to rat and human data. **(a)** For this data from [17], a 2AFC delayed response task was used in which the subject experiences an auditory stimulus (Tone A) of a particular amplitude, a delay period, a second auditory stimulus (Tone B) of a different amplitude, and finally the choice to go either left or right. If Tone A was louder than Tone B, then a rightward choice receives a reward, and vice-versa. **(b)** The psychometric weights recovered from the first 20,000 trials of a rat. Weights in the legend labeled with a "-1" superscript indicate that the weight carries information from the previous trial. The faded vertical gray lines indicate session boundaries. In addition to being fit with its own trial-to-trial volatility hyperparameter $\sigma_k$, each weight is also fit with an additional hyperparameter $\sigma_{k,\text{day}}$ for volatility between sessions. This results in "steps" at the session boundaries for some weights (see Sec. 3.3). Each weight also has a 95% posterior credible interval, indicated by the shaded region of matching color (for derivation refer to Sec. S2). **(c)** The psychometric weights recovered from a human subject.

subjects to uncover how behavior evolved in this particular task. Here we show examples from one rat and one human subject (Figs. 3b,c); see Figs. S2 & S3 for analysis of additional rats and human subjects.

## 5.1 Rat data

Behavior is highly dynamic in the case of a rat (Fig. 3b), reflective of the animal's initial uncertainty about the task structure and gradual honing of its behavioral strategy. First, we notice that the animal starts naive: the initial strategy does not depend upon the two auditory stimuli at all, as both the Tone A & B weights (red & yellow) begin near 0. Instead, behavior is clearly influenced by the previous trial: the weights on answer history (purple; preference to choose the side that was correct on the previous trial, or "win-stay/lose-switch") and on choice history (green; preference to choose the same side as on the previous trial, or "perseverance") both dominate initially. There is also an overall tendency to choose left, as indicated by the negative bias weight (blue). As training progresses, both the bias and dependencies on task history steadily decrease, suggesting that the rat is learning the task structure.

Second, we can compare the evolution of the weights on Tone A vs. Tone B. The sensitivity to the value of Tone B is developed very early in training, and quickly grows to a large negative value (preference to go left when Tone B is loud). In contrast, the sensitivity to Tone A stays close to zero for many thousands of trials before growing to have a large positive value (preference to go right

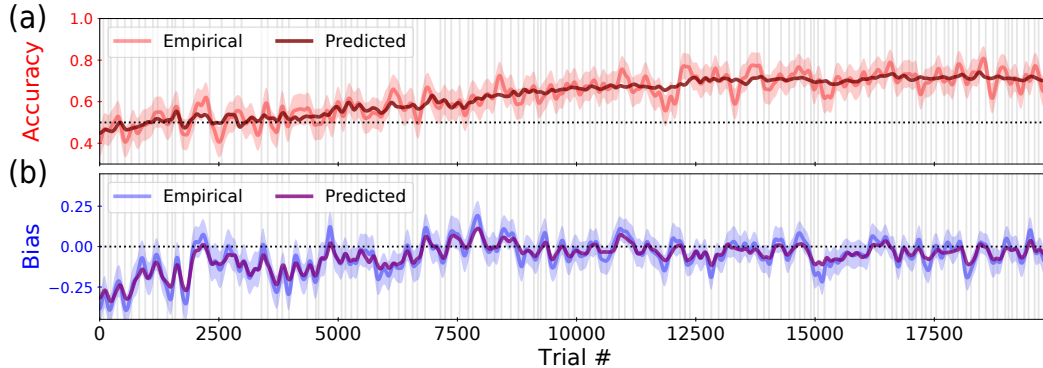

Figure 4: Comparing to empirical metrics. **(a)** The empirical accuracy of the rat in red, with a 95% confidence interval indicated by the shaded region. We overlay the predicted accuracy from model weights in maroon, using $P(\mathrm{Correct})$ for each trial instead of the empirical $\{0, 1\}$. **(b)** The empirical bias of the rat, represented as the correct side minus the animal's choice for each trial, where {Left, Right} = {0,1}. We plot a 95% confidence interval indicated by the shaded region, as well as the predicted bias from model weights, substituting $P(\mathrm{Right})$ for the animal's choice. All lines are smoothed with a Gaussian kernel of $\sigma = 50$. Predicted performance and bias are calculated using cross-validated weights (calculations and cross-validation procedure detailed in Secs. S3 & S4).

when Tone A is loud). Again, this observation is consistent with the intuition that associative learning is stronger for the most recent stimulus. The temporal separation of Tone A from the choice not only makes it more difficult to learn the association, but also makes leveraging knowledge of that association more difficult since the rat must work to maintain information about Tone A in working memory [18, 17]. Despite this, we see that the animal ultimately develops weights of equal magnitude and opposite signs for the two stimuli, again demonstrating successful learning.

Finally, we observe a small but significant sensitivity to the previous trial's stimuli (pink); the positive value indicates a preference to go right when the average of Tones A & B on the previous trial was higher. This reconfirms the dependence of choice behavior on sensory history found in [17].

## 5.2 Human subjects

In contrast to the rat, the weight trajectories for the human subject are largely stable and reflect accurate behavioral performance (Fig. 3c); not much learning is happening. This is expected, as a human subject can understand the task structure and execute the correct behavioral strategy from the very first trial. We emphasize that the strength of our model is not only its flexibility to fit the dynamic behavior of the rat, but also to automatically detect and confirm the stable behavior of the human. While the human dataset is stable enough to be fitted using standard logistic regression, it would require starting from the assumption that behavior was indeed stable.

Our method also allows several interesting observations regarding the types of decision-making biases a human subject might possess. For example, there is a non-zero choice bias (blue) with a slow fluctuation, that tends leftward over most of the session. Also, while the weights for Tones A & B are clearly the two largest, the magnitude of the Tone B weight is consistently larger, indicating a higher sensitivity to the more recent stimulus. Furthermore, the weight on sensory history (pink) is non-vanishing, once again corroborating the findings of [17]; whereas behavior was even better explained without the weights on answer and choice history (see Sec. S1 for more detail).

## 5.3 Comparison to conventional measures

Finally, we ask how well our model actually describes the animal's choice behavior. To this end, we relate our model back to more conventional measures of behavior, considering two important measures most commonly used by a trainer: the empirical accuracy (Fig. 4a) and the empirical bias (Fig. 4b). The empirical accuracy tracks the local fraction of trials with a correct response, whereas the empirical bias captures the tendency to prefer one side on error trials (see Sec. S3 for details). The two measures were then compared with the corresponding quantities predicted by our

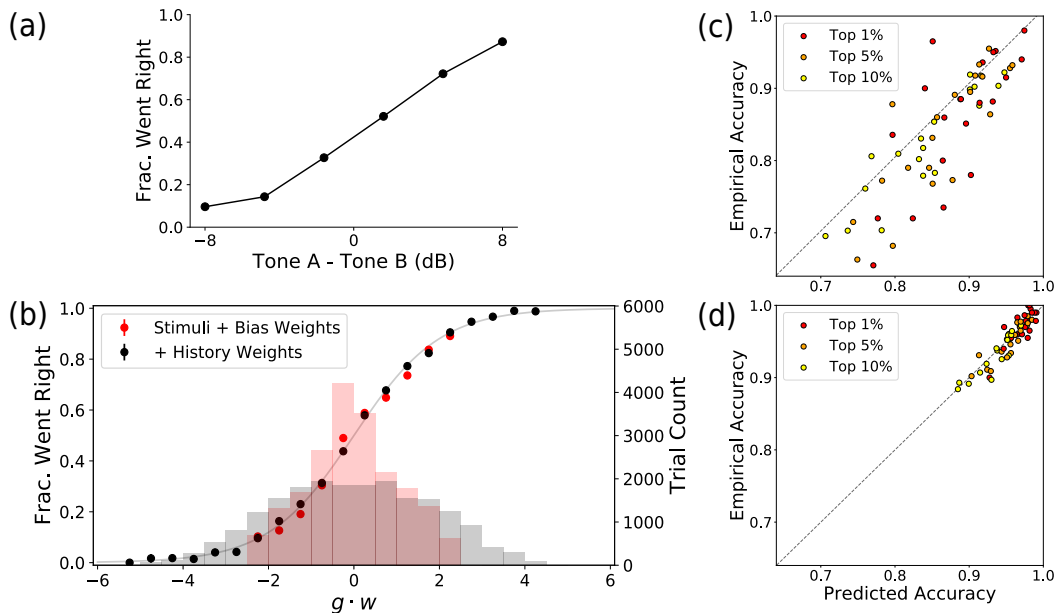

Figure 5: Psychometric curve and model predictions. **(a)** A conventional psychometric curve for a rat, generated from a subset of trials at the end of training where Tone B is held constant. **(b)** The same rat during early training, with two models: the basic model with stimuli and bias weights only (red), and the history-aware model (black). The histograms (with right-side axis) show the number of trials within each $\mathbf{g} \cdot \mathbf{w}$ bin. The dots (with left-side axis) plot the fraction of trials, within each bin, in which the rat went right ($y_t = 1$). **(c)** For all 20 rats in the population, we plot the predicted accuracy vs. empirical accuracy for the top 1, 5, & 10% of most strongly predicted trials in the basic model (red). **(d)** Same as (c) but for the history-aware model (black).

model. The close match between empirical and predicted performance validates the model's ability to capture the animal's true dynamic strategy, in addition to the already-demonstrated success of our inference method to find the best weights and hyperparameters given the model. It also emphasizes that our analysis provides highly interpretable measures that could successfully replace (and extend) conventional training evaluators.

## 6 Exploring lapse

Looking at the psychometric curve of a single rat from its end-of-training data (Fig. 5a), it is clear that the rat does not achieve perfect performance even on the easiest trials (left/right ends of the stimulus axis). This gap in performance is particularly common in rodent data and there is much speculation as to its cause [19]. One possible hypothesis is that the trials are not easy enough, and that perfect performance would be achieved on sufficiently easy trials; in other words, this is to explain the gap as a result of insufficient sensitivity to task stimuli [20–22]. An alternative hypothesis is that there exists a so-called "lapse rate" inherent in the animal's behavior, for example as an effect from an $\epsilon$-greedy strategy where the animal makes a completely random choice on a certain fraction of trials, perhaps for exploratory purposes. Our analysis of the rat data can provide an answer to the debate, as it captures the behavior of the animal precisely enough to *predict*, not just describe.

To explore the predictive power of our method further, we look at two distinct models in Fig. 5b: the basic model (in red) has dynamic weights only on the task stimuli (Tones A & B) and choice bias, while the history-aware model (in black) has additional weights for various history dependencies. On the x-axis of Fig. 5b, we have binned all trials of our rat according to their $\mathbf{g}_t \cdot \mathbf{w}_t$ values. Recall that in our model, larger magnitudes of $\mathbf{g} \cdot \mathbf{w}$ result in more confident predictions, with predicted choice probabilities closer to 0 or 1. We see that the empirical probability of choosing right within each bin of $\mathbf{g} \cdot \mathbf{w}$ (plotted in dots) matches the predicted probability according to the logistic function of Eq. 1 (plotted as faded gray curve). We then plotted the number of trials in each $\mathbf{g} \cdot \mathbf{w}$ bin in the histograms

(with right-side axis). We see that for our basic model, the trial predictions are never more confident than 90% (no tails on the red histogram), whereas our history-aware model has a substantial portion of trials predicted with almost 100% confidence (longer tails on the black histogram). In terms of the cross-validated log-likelihood, the history-aware model provides a 20% boost over the basic model. All model predictions are calculated on held-out data; see Sec. S4 for details.

Finally, we directly compare the model-predicted accuracy to the empirical accuracy, across all 20 rats, for both the basic model (Fig. 5c) and the history-aware model (Fig. 5d). Sorting trials by their predicted accuracy, we plot the top 1, 5, & 10% of trials in each rat and find that almost all rats have a significant proportion of their trials being predicted with >95% accuracy in the history-aware model (Fig. 5d).

We thus demonstrated that our method can predict the rats' choice behavior with near-perfect accuracy on a significant subset of trials. This finding contradicts the hypothesis postulating an inherent lapse rate, where the animal is making random choices on a subset of trials (where such randomness would prevent prediction above a certain accuracy). While random choice is a well-established behavioral strategy seen in many experimental settings, our method allows for critical disambiguation between true randomness and deterministic strategies that may appear random [23]. Our method, with the full history-aware model, is able to quantify and explain gaps in performance typically left unexplained by conventional analyses.

# 7 Discussion

We presented a method for efficiently and flexibly characterizing the dynamics of psychophysical behavior, allowing for unprecedented insight into how animals learn new tasks. We have made key advancements with regard to both efficiency and scalability, allowing us to quickly fit a complex, trial-to-trial description of behavior for even the largest of datasets. We demonstrated on a real dataset (of unusually large size) the explanatory as well as predictive power of our method, as compared to two conventional measures of behavior. In particular, the flexibility of the model allowed us to address an important open question in behavioral psychology, known as lapse.

Our approach is developed under a simple generic model of psychometric behavior, which worked nicely for the datasets we analyzed in this paper. Here we briefly discuss two aspects of the model that may be extended in a future study, potentially to address specific features of different tasks. First, while the weight trajectories are allowed to evolve over time, the volatility hyperparameter $\sigma$ is a single value optimized over the entire dataset. When analyzing a long trajectory, it may be necessary to also allow $\sigma$ to slowly vary over time, so that the dynamics of early training and the stability of late training can be explained separately. Including more complex parameterizations of the prior, such as the overnight $\sigma_{\text{day}}$ described in Sec. 3.3, may also provide a practical solution in modeling sudden, step-like changes in behavior. Second, the success of our method is fundamentally dependent on the ability of the psychometric model to correctly describe the animal's behavior. Different tasks may require more careful modeling of certain aspects of the choice behavior. In particular, our model only applies to 2-alternative forced choice tasks in its current form, though there is a clear extension to multi-alternative choice [7]. Despite these limitations, we expect the agnostic flexibility, explanatory power, and computational efficiency of our method to make it a useful tool for exploring behavioral dynamics. Our Python package PsyTrack should make the analysis easily accessible [6].

Our method can be easily applied to the vast troves of largely unanalyzed animal training data to provide both scientific insight and practical utility. The immediate applicability, as a potential everyday tool for scientists-trainers, places our method a significant step forward from previous works that offered theoretical paths [16]. At the lowest level, this method allows trainers to stay aware of the behavioral strategies developed by their animals, useful for identifying common pitfalls and disentangling distinct strategies that may appear similar on the surface. Furthermore, while many trainers are already using various automated heuristics during training, the output of our method can be used as a more specific and accurate input to such heuristics. By enabling a quantitative feedback loop where the trainer can (i) diagnose a problem, (ii) prescribe an adaptively optimized training program to correct it, and (iii) monitor the consequence of that correction, we feel that our method will set a new standard for systematic animal training.

**Acknowledgements**

This work was supported by grants from the Simons Foundation (SCGB AWD1004351 and AWD543027), the NIH (R01EY017366, R01NS104899) and a U19 NIH-NINDS BRAIN Initiative Award (NS104648-01).

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
