[Supplementary Material]

# Supplementary Material

## Efficient inference for time-varying behavior during learning
### N.A. Roy, J.H. Bak, A. Akrami, C.D. Brody, J.W. Pillow

Here we present more details of our analysis. This document contains four sections and three figures:

### S1    Variable standardization, model selection, and non-identifiability

It is important that the variables used in $\mathbf{g}$ are standardized such that the magnitudes of different weights are comparable. For continuous variables (Stimuli A & B), we normalize the values such that they reflect a standard normal distribution (i.e., subtract the mean and divide by the standard deviation). For categorical variables (previous answer, history, reward dependencies), we constrain values to be $\{-1, 0, +1\}$. Specifically, the answer variable is coded as a -1 if the correct answer on the previous trial was left, +1 if right; the choice variable is -1 if the animal's choice on the previous trial was left, +1 if right; and the reward variable is -1 if a reward was not received on the previous trial, +1 if a reward was received. For these variables depending on the previous trial, they are set to 0 if the previous trial was a mistrial. Mistrials (instances where the animal did not complete the trial, e.g., by breaking center fixation before the end of the trial) are otherwise omitted from the analysis. The choice bias is fixed to be a constant +1.

The decision as to what variables to include when modeling a particular data set can be determined solely using the log-evidence as discussed in Sec. 3.2 — the model with the highest log-evidence is considered to be best (though this comparison could also be swapped with a more expensive comparison of cross-validated log-likelihood, see Sec. S4 and Fig. S1). Unfortunately, there is no way to make this comparison between models without first running a model with every combination of variables and choosing after the fact. However, the fitting can be parallelized across models (i.e., with 3 optional history variables, there are 8 possible models, which can each be fit in parallel).

Finally, we address a concern about non-identifiability in our model. This occurs if one variable is a linear combination of some subset of other variables, in which case there are many weight values that all correspond to a single identical model. Fortunately, the posterior confidence intervals discussed in Sec. S2 will indicate that a model is in a non-identifiable regime — since the weights can take a wide range of values to represent the same model, the confidence intervals will be very large on the weights creating the non-identifiability.

### S2    Calculation of posterior credible intervals

In order to estimate the extent to which our recovered weights $\mathbf{w}_{\mathrm{MAP}}$ are constrained by the data, we adapted a fast method for inverting block-tridiagonal matrices to calculate central blocks of the inverse of our (extremely large) Hessian, providing a Gaussian approximate marginal posterior over the time-varying weight trajectories (as shown in Fig. 3). The algorithm is taken from Appendix B of [S2] which discusses calculating the diagonal elements of the inverse of a tridiagonal matrix, then describes how the approach can be generalized to block-tridiagonal matrices. The algorithm requires order $NK^3$ scalar operations for calculating the central blocks of our inverse Hessian. If $H$ is the Hessian matrix of our weights corresponding to the model with the highest log-evidence (returned at the end of the optimization procedure outlined in Algorithm 1 in the main text), then this calculation yields:

$$A = \mathrm{diag}(H^{-1}) \tag{S1}$$

Using the diagonal of the inverse Hessian, we can take $\sqrt{A}$ to estimate a one standard deviation interval on either side of each weight on every trial. By using two standard deviations, we approximate the 95% posterior credible interval shown throughout the paper.

### S3    Calculation of empirical performance measures

Here we explain how the two empirical performance measures (shown in Fig. 4 in the main text) were calculated.

The first measure, accuracy, is *empirically* calculated by constructing a vector of length $N$ where the $i^{\mathrm{th}}$ entry is a 0 if the animal answered incorrectly on that trial and a 1 if answered correctly. This vector is then smoothed (we use a Gaussian kernel of $\sigma = 50$) to get the empirical accuracy plotted in red in Fig. 4a. To calculate the *predicted* accuracy, we use the cross-validated weights recovered for each trial $\mathbf{w}_t$ (cross-validation procedure detailed below in Sec. S4) and the inputs on that trial $\mathbf{g}_t$ to calculate $P(\mathrm{Right})$ (see Eq. 1); from that we then use the known correct answers to calculate $P(\mathrm{Correct})$ for each trial which we then smooth with the same Gaussian kernel to get the maroon line in Fig. 4a.

One of the strengths of our model is the ability to isolate a specific bias weight (blue in Figs. 3b+c) that is distinct from other, potentially confounding aspects of behavior (e.g., increased sensitivity to a stimulus or a choice history

dependency). What is often measured instead is simply a moving average of the fraction of trials where the animal choose to go left (or right). Because training regimens do not always present left and right stimuli in equal proportion, we instead define empirical bias here as a preference for left or right only on *incorrect* trials.

Specifically, we construct a vector of length $N$ where the $i^{\text{th}}$ entry is the animal's choice minus the correct answer, where both choice and answer are coded such that "left" is 0 and "right" is 1. Thus, *empirical* bias is on each trial one of $\{-1, 0, +1\}$. This vector is then smoothed (we use a Gaussian kernel of $\sigma = 50$) to get the empirical bias plotted in blue in Fig. 4b. We calculate the *predicted* bias in a similar manner to the predicted performance, using cross-validated weights to calculate $P(\text{Right})$ for each trial and substituting that value in for the animal's choice — thus for each trial we get a value from $(-1, +1)$ which we then smooth with the same Gaussian kernel to get the purple line in Fig. 4b.

## S4 Cross-validation

When making predictions about specific trials, the model should not be trained using those trials. We implement a 10-fold cross-validation procedure where the model uses a training set composed of a random 90% of trials at a time. We modify the prior $\Sigma$ discussed in Sec. 3.1 such that the gaps created by removing the test set are taken into account. For example, if trial $t$ is in the test set and trials $t-1$ and $t+1$ are in the training set, then we modify the value on the diagonal of $\Sigma$ corresponding to trial $t-1$ from $\sigma^2$ to $2\sigma^2$ to account for the missing entry in $\Sigma$ created by omitting trial $t$ from the training set. To predict the animal's weights at a trial $t$, we use the cross-validation fold where $t$ is in the 10% test set, and approximate $\mathbf{w}_t$ by linearly interpolating from nearest adjacent trials in the training set. Thus, we can infer a set of cross-validated weights for every trial from which we can use the carrier vector $\mathbf{g}_t$ for that trial to get a cross-validated $P(\text{Right})$.

---

**Figure S1. Hyperparameter optimization**   Our decoupled Laplace method directly searches the hyperparameter space to find the best set of values where the evidence (or the marginal likelihood) is maximized. In order to show the efficacy of our method, we confirm with a grid search that the algorithm converges on the hyperparameters with the highest evidence and highest cross-validated log-likelihood.

**Figure S2. Rat data**   Extending from Figure 3b, we show results from an additional 8 rats not shown in the main text. (data from [S1])

**Figure S3. Human data**   Extending from Figure 3c, we show results from the remaining 8 human subjects not shown in the main text. (data from [S1])

---

**Figure S1: Hyperparameter optimization.** Here we simulate two weights and search the 2-dimensional hyperparameter space of values for $\sigma_1$ and $\sigma_2$. Dots indicate the hyperparameter values recovered using our decoupled Laplace algorithm, from 10 individual behavioral realizations (light blue dots, as in Fig. 2a) and averaged over all realizations (dark blue dot with cross). We confirm that these are near the optimal pair of hyperparameters by comparing to **(a)** the average log-evidence, and **(b)** the average cross-validated log-likelihood calculated on a fixed grid of hyperparameters values, shown in gray-scale.

**Figure S2: Rat data.** The psychometric weights recovered from the first 20,000 trials of 8 rats, as in Figure 3b. Also included for each subject are the comparisons to conventional behavioral measures, as in Fig. 4.

**Figure S3: Human data.** The psychometric weights recovered from each of the human subjects, as in Fig. 3c. Also included for each subject are the comparisons to conventional behavioral measures, as in Fig. 4. Results for subject #6 were shown in the main text, and are omitted here.