[Reviews · NeurIPS 2018]

Reviewer 1



The paper presents a new estimator for a dynamic logistic regression model used to characterize human or animal behavior in the context of binary decisions tasks. Despite the very large number of parameters such a model can be estimated robustly and with tolerable computational cost by exploiting the structure of the posterior covariance. The estimator is then applied to a delayed auditory discrimination task in humans and animals. The results show signatures of learning and history interference in the rodent compared to human. Additionally the model fit manages to predict behavior in test data. The model is simple and interpretable which will likely make the use of the tools popular within the community. Comments: - the choice of prior for w is a bit odd - since the variance of the marginal for w_t grows linearly with t; Also the fact that the noise is independent seems somewhat limiting - how about things like fluctuation in attention those would likely affect the processing of both stimuli and potentially even some of the history... - it would be nice if you would directly compare the dynamics of w with the dynamics of learning, as measured behaviorally; the prediction estimates in fig 3 goes into that direction but it would be interesting to see a direct visualization of the two. - similarly do the fluctuations in ws predict fluctuations in performance in humans? - it wasn't clear if the code associated will be released with the paper Minor: - unspecified w_0: I am assuming w_0 = 0 L should be defined somewhere explicitly in eq 3 - unclear what 'behavioral realization' is supposed to mean in relation to figure 2: are individual lines MAP estimates for w, where the true w is kept fixed but the observation noise is redrawn independently? And are the hyperparameters estimated in panel b the same way or are the ws redrawn for each experiment? -at some point the text mentions sparse priors for w which could be misleading - it's just the prior covariance matrix is sparse. Post rebuttal: Thanks for the clarifications. I still the prior should be justified by more than mathematical convenience but it's ok as long as the text justifies it accordingly.

Reviewer 2



This is an impressive paper that effectively tackles one of the most important (yet far from the most studied) challenges in neuroscience - characterising complex phenotypes (Krakauer et al., Neuron 2017). Unlike model-based studies (e.g. Corrado & Doya, JoN 2007; Luksys & Sandi, CON 2011; Nassar & Frank, COBS 2016), which focus on employing models to gain insights into animal behaviour, this study employs the model-free approach that perhaps is less insightful in each specific case but can be easily adapted to a multitude of behavioural scenarios and hence can be used much more easily by cognitive and behavioural neuroscientists, and may contribute significantly to understanding behaviour and refining the experiments. Although it's a strong paper and a clear accept even in the current form, it could be improved further by addressing the following points: - more specific references to behavioural studies where it's particularly relevant could be provided (e.g. paragraph 1 makes very good points but no references), especially given that only one type of experiment is analysed in the paper (which is understandable due to space limitations). - it would also be helpful to mention model-based analysis studies and briefly compare and contrast the approaches. This may also provide a good base for discussing limitations, as for many learning and decision making studies (e.g. Morris water maze) behaviours cannot be directly related to sensory modalities, hence it's unclear how applicable the presented approach can be - how is simulated data in section 4 created? - although I agree that history can account for part of perceived randomness in actions (as the authors showed in Fig. 4) it's important to emphasize that exploration/randomness is a well established phenomenon (which may be more prominent under low stress), and only in this particular task it doesn't seem to play a major role (although is still non-zero) - I wonder if discrepancy in tone A and B weights in human data (first steady, 2nd noisy) may be some fitting artefact. --- Thanks to the authors for their responses. It's a nice paper - clear accept, and I look forward to reading the final version.

Reviewer 3



This paper aims to quantify behavior during training (as opposed to after training) of animals and humans via a dynamic logistic regression model that tracks trial-to-trial changes. This approach scales to large behavioral datasets (requiring ~500K parameters) by using the decoupled Laplace approximation. The advantages of this method include being able to study how various task factors (such as stimuli, choice history, etc) impact learning on the single trial level. I really like the motivation for this paper. It’s true that often in neuroscience and cognitive science experiments we neglect the dynamics of learning the task and only look at steady state behavior. This approach could point to new theories for learning and behavioral strategies, so the significance is high. The method also appears to be well-validated via simulation in section 4 and figure 2. However a large drawback to this paper is its severe lack of background literature review. How does this approach relate to for instance beta series analysis? How does it compare against simply blocking the data into eg 10 trial blocks and performing normal logistic regression? (Is the issue overfitting?) What is the relative computational/time complexity of blocking trials vs their approach vs not using the decoupled Laplace approximation? Are there instances in which the evolution of weights cannot be assumed to evolve with independent Gaussian noise? Finally, I would have liked to have seen this method applied to a much more complex task. For the humans especially this seems too easy, and as the authors even comment, humans don’t really display learning since they can understand the task structure from instructions. Looking at the results for all humans in the supplement, many of them don’t have evolving weights at all, making this task quite unsuitable for the motivations given in the intro. Update after reading rebuttal: The authors' response adequately addressed all my concerns, and I look forward to seeing the follow-up to this work on more complex tasks!